# Insect Herbivory on Main Stem Enhances Induced Defense of Primary Tillers in Rice (*Oryza sativa* L.)

**DOI:** 10.3390/plants12051199

**Published:** 2023-03-06

**Authors:** Lu Tong, Wanghui Wu, Yibin Lin, Daoqian Chen, Rensen Zeng, Long Lu, Yuanyuan Song

**Affiliations:** 1State Key Laboratory of Ecological Pest Control for Fujian and Taiwan Crops, Key Laboratory of Ministry of Education for Genetics, Breeding and Multiple Utilization of Crops, College of Agriculture, Fujian Agriculture and Forestry University, Fuzhou 350002, China; 2Guangxi Zhuang Autonomous Region Forest Inventory & Planning Institute, Nanning 530022, China; 3College of Natural Resources and Environment, South China Agricultural University, Guangzhou 510642, China

**Keywords:** clonal plants, physiological integration, induced defense, interplant communication, jasmonic acid signaling

## Abstract

Clonal plants are interconnected to form clonal plant networks with physiological integration, enabling the reassignment as well as sharing of resources among the members. The systemic induction of antiherbivore resistance via clonal integration may frequently operate in the networks. Here, we used an important food crop rice (*Oryza sativa*), and its destructive pest rice leaffolder (LF; *Cnaphalocrocis medinalis*) as a model to examine defense communication between the main stem and clonal tillers. LF infestation and MeJA pretreatment on the main stem for two days reduced the weight gain of LF larvae fed on the corresponding primary tillers by 44.5% and 29.0%, respectively. LF infestation and MeJA pretreatment on the main stem also enhanced antiherbivore defense responses in primary tillers: increased levels of a trypsin protease inhibitor, putative defensive enzymes, and jasmonic acid (JA), a key signaling compound involved in antiherbivore induced defenses; strong induction of genes encoding JA biosynthesis and perception; and rapid activation of JA pathway. However, in a JA perception OsCOI RNAi line, LF infestation on main stem showed no or minor effects on antiherbivore defense responses in primary tillers. Our work demonstrates that systemic antiherbivore defense operate in the clonal network of rice plants and JA signaling plays a crucial role in mediating defense communication between main stem and tillers in rice plants. Our findings provide a theoretical basis for the ecological control of pests by using the systemic resistance of cloned plants themselves.

## 1. Introduction

Plants provide basic organic nutrients for various herbivores on the planet. They are constantly exposed to insect herbivores, which can cause severe damage to crop production. In nature, plants have developed both constitutive and induced defenses against insect herbivores [1]. Constitutive defense is present at any time. Induced defense refers to defense responses activated by attackers that render increased plant resistance to subsequent attacks. Induced defense allocate resource and energy to defense only when needed. It also prevents autotoxicity of the accumulation of defensive compounds [2]. The induced defense should be favored over long-standing, constitutive defense since defenses are costly and not needed all the time. Induced defense is widespread in plants and can exert strong effects on herbivore populations and communities [3,4].

Induced plant defense is primarily regulated by three signaling hormones: jasmonic acid (JA), salicylic acid (SA), and ethylene (ET). These signaling pathways either synergistically or antagonistically interact with each other to fine-tune the plant defense response depending on the type of attacker encountered [5]. For plant responses to chewing insects, the JA signaling pathway plays a vital role in induced defense [6,7]. Upon insect herbivory, plants launch defense responses aided by the recognition of herbivore-associated molecular patterns [8], followed by the activation of a sophisticated regulatory network involving Ca^2+^, reactive oxygen species (ROS), mitogen-activated protein kinase (MAPK) cascades, jasmonic acid (JA) signaling, the induction of defense-related genes, and increased biosynthesis of defense compounds [9].

Herbivore-induced plant defense occurs not only locally, but also systematically throughout the entire plant individual. A localized herbivore attack can further lead to systemically induced defense [10], leading to systemic alteration of toxic secondary metabolites, protease inhibitors, physical barriers, and tolerance traits at the whole plant level for plant protection against herbivores [11]. It has been demonstrated that JA signaling plays an essential role in plant systemic-induced defense against herbivores [12,13,14,15].

Upon insect herbivory how plants perceive the feeding cues and communicate those perceptions are vital for their survival. Plants can gain early warning signals via plant–plant communications [16,17]. They can communicate with each other via either herbivore-induced plant volatiles (HIPVs) [18], underground common mycorrhizal networks [19,20], or parasitic plant [21].

Some plants vegetatively generate many genetically identical but independent individuals (ramets or clones) via spacers (rhizomes and stolons) [22] to form a clonal plant network. These clonal plants are functionally independent plants but physically connected by stolon or rhizome internodes. Physical connections in the clonal network allow clonal plants to share essential resources such as photosynthates, nutrients, and water within clones [23,24], facilitating alleviation of abiotic stress in heterogeneous environments. Nutrients and water can be translocated from clones growing in nutrient-rich and well-watered soils to clones growing in stressed soil. The clonal plants in the network become physiologically integrated. Physiological integration, a unique feature of most clonal plants, is considered to contribute to the tremendous vitality of clonal plants in nature, their dominance in many ecosystems, and the invasion success of alien clonal plants [25,26,27,28]. Clonal plants play a crucial role in maintaining plant productivity and ecosystem stability [29,30].

Many studies have documented the movement and sharing of nutrients among the members of the clonal plant network, while much less is known about defense communication within the clones [31,32]. Physical connections make it possible to easily share defense signals among individuals in the clonal networks. The systemic induction of antiherbivore resistance via clonal integration may frequently operate. When a member is infested by herbivores other members in the clonal network can quickly gain early warning signals to activate their defense responses for impending herbivore attacks. Although a few studies have shown that defense communication exists in clonal networks, most of these studies focused mainly on dicotyledonous and stoloniferous plants such as *Trifolium repens* [33] and *Alternanthera philoxeroides* [34], relatively little attention has been paid to monocotyledonous tillering plants and important crops.

Rice (*Oryza sativa* L.) is one of the most important food crops and a typical clonal plant, which produces tillers (clonal plants) from the nodes of the basal non-elongated internodes of the main stem (culm) [35]. Tillering capacity is one of the most important traits of plant architecture and grain yield. A normal mature rice plant has a main stem and a number of tillers. Primary tillers are developed at the shoot apex of the main stem, and secondary tillers are developed from primary tillers. We hypothesized that leaf herbivory on the main stem will increase antiherbivore resistance and defense responses in primary tillers and hormonal signals play a crucial role in defense communication between the main stem and its primary tillers. In this study, we examined the effects of leaf damage of the main stem by the rice leaffolder (LF; *Cnaphalocrocis medinalis*) (Lepidoptera: Pyralidae), a major insect pest of rice, and exogenous application of methyl jasmonate (MeJA) on antiherbivore resistance, JA signaling, protease inhibitor level and defense-related enzyme activities in the primary tillers. To determine the role of JA signaling the transgenic rice plants OsCOI1-RNAi deficient in JA perception generated via RNAi were used to investigate resulting changes in anti-LF resistance and defense responses in the primary tillers.

## 2. Materials and Methods

### 2.1. Plant Growth

Seeds of rice (*Oryza sativa* L. ssp. Japonica cv. Ishikari-shiroke) were sterilized with 10% H_2_O_2_ for 10 min, washed with distilled water, and germinated at 25 °C for 48 h. The uniform germinated seeds were sown in a tray with length of 56 cm × width 37 cm × height of 8.5 cm and grown in a greenhouse (28 ± 2 °C, 80% relative humidity, 12 L:12 D) for 15 d. Then, four seedlings were transplanted into a plastic bucket (26 cm in diameter and 17 cm in height). The seedlings were watered every day and applied with 200 mL of compound fertilizer solution (concentration: l g/L, N: P_2_O_5_: K_2_O = 1:1:1) every week. Forty days after transplanting, the plants were used for experiments. Each plant had 3–4 primary tillers.

To evaluate possible role of the JA signaling in defense communication between main stem and primary tillers, RNAi was used to silence the expression of rice *OsCOI1* (GenBank accession no. DQ028826) gene. The OsCOI1 RNAi line was obtained as described by Ye et al. [36]. Silencing *OsCOI1* reduced plant resistance to rice leaffolder and compromised rice-induced defense. The corresponding wild-type (WT) was Ishikari-shiroke (*Oryza sativa* L. ssp. Japonica).

### 2.2. LF Insects

Rice leaffolder (LF; *Cnaphalocrocis medinalis*) caterpillars used for insect herbivory was collected from the rice field on the campus of South China Agricultural University in Guangzhou (China) and maintained on rice seedlings (cv. Ishikari-shiroke) in the growth chamber at 28 ± 2 °C, 12 h:12 h light/dark regimen, and 80% relative humidity. The uniform third-instar larvae were used for all bioassays described.

### 2.3. Experimental Design

In all experiments, rice plants of OsCOI1 RNAi line and corresponding WT were simultaneously used for bioassay and biochemical and molecular assay. Each phenotype received four treatments: (1) Control, both main stem and primary tiller did not receive any treatment; (2) non-induced + LF, main stem did not receive any treatment, but primary tiller received LF infestation; (3) LF induced + LF, main stem was inoculated with LF larvae for two days and then the primary tiller was inoculated with LF larvae; (4) MeJA induced + LF, main stem was sprayed with MeJA for two days and then the primary tiller was inoculated with LF larvae.

For LF-induced + LF treatment, the first and second leaves (from the top) of the main stem of each rice plant were inoculated with two third-instar LF larvae that had been starved for 2 h. Then the main stem was covered with gauze bags to prevent the insects from moving to the tillers. Two days after insect inoculation, the larvae were removed, and then the primary tillers were inoculated with two third-instar LF larvae that had been starved for 2 h. The leaves of damaged primary tillers were sampled at 0, 1, 3, 6, 12, 24, and 48 h after insect inoculation on primary tillers. The third leaf of the primary tiller infested by LF was harvested and frozen in liquid nitrogen and stored in a refrigerator at −80 °C. The samples were used to detect JA contents, defense-related enzyme activities and gene transcript levels, and trypsin inhibitor content. Each treatment had four replicates.

For MeJA + LF treatment, the leaves of the main stem were sprayed with 1 mM MeJA solution (prepared with 50 mM phosphate buffer, adjusted to pH 8.0, with 0.01% Tween 20), and then covered with a sealed bag for 2 d. During the treatment period, the plants were placed in a ventilated and normal light environment, and the bags were opened quickly every 3 h for air exchange and then sealed. The rice plants sprayed with the buffer solution on the main stem and primary tiller without LF inoculation (control), and the rice plants sprayed with the buffer solution on the main stem and primary tiller inoculated with LF larvae (non-induced + LF) served as two controls. To avoid airborne plant volatile-mediated interplant communication, fans always run to disperse the plant volatiles.

### 2.4. LF Bioassays

Uniform third-instar LF larvae that had been starved for 2 h were weighed. One larva was inoculated on the second leaf of primary tiller with the main stem that had been either inoculated with LF larvae for 2 d, treated with 1 mM MeJA, or treated with the buffer solution for 2 d. Then, larva from each plant was collected and weighed. LF mass gain was calculated as increased percentage (%). Each treatment had 20 replicates.

### 2.5. Jasmonic Acid (JA) Analyses

The leaves of LF-damaged primary tillers were sampled at 0, 1, 3, 6, 12, 24, and 48 h after insect inoculation on primary tillers. JA content was quantified by GC analyses with external JA standards (Sigma-Aldrich, St. Louis, MO, USA) using the method described by Song et al. [20]. The leaf sample (0.1 g) of LF-damaged primary tillers was prepared as described above in the section of experiment design, ground in liquid nitrogen, and transferred to 2 mL of centrifuge tube. Then, 1 mL mixture of acetone/50 mM citric acid (*V*/*V* = 7/3) was added, followed by 1 mL ethyl acetate, then mixed on the oscillator. The extract was centrifuged at 14,000× *g* for 15 min at 4 °C and the supernatant was dried by N_2_. The obtained supernatant was methylated with trimethylsilyldiazomethane. The resulting volatiles were collected by headspace solid-phase microextraction (HS-SPME) on Tenax adsorbents and eluted with n-hexane. The elute was detected by gas chromatography with hydrogen ion flame detector (FID) as described by Ye et al. [36]. Standard MeJA (18 mg/mL) (Sigma-Aldrich, St. Louis, MO, USA) was used to confirm the recovery rate of JA.

### 2.6. TrypPI Analyses

TrypPI activity was evaluated by a colorimetric method using the protein chromophore azocasein as a substrate [37]. Leaf tissue of the primary tillers (0.1 g) was ground in Tris-HCl buffer (0.2 M, pH 8.0) containing 0.1% Tween 20, then centrifuged at 12,000× *g* for 20 min at 4 °C. The obtained supernatant was used for analysis. Plant extract (200 mL) and 0.1 mg/mL trypsin (600 mL) were mixed in each reaction and incubated at room temperature for 10 min, then mixed with 100 mL of 25 mg/mL azocasein, incubated at 37 °C for 40 min. The resulting solution was centrifuged at 12,000× *g* for 10 min, the obtained supernatant (200 mL) was mixed with 200 mL of 0.5 M NaOH, and its absorbance was measured at 450 nm. The protease inhibitor levels were calculated by using a standard curve and expressed as nM protease inhibitor per mg protein. The protein content was measured using the Bradford assay with BSA as standard [38].

### 2.7. Quantitative Real-Time PCR Analysis

Differential expression of key genes in JA pathway and Bowman−Birk protease inhibitor gene (*OsBBPI*) was quantified by real-time qRT-PCR using the RNA samples isolated from rice leaves of primary tillers gained from different treatments. The actin gene served as a reference gene. Total RNA from rice leaves was extracted using the method described by Kiefer [39]. RNA extraction and reverse transcription of plant samples were conducted using the method described by Ye et al. [40]. Real-time PCR was performed according to the procedure of UltraSYBR two-step fluorescence quantitative PCR kit (ComWin Biotech, Beijing, China). The gene-specific primer sequences used are listed in Appendix A. Reaction conditions for thermal cycling were initial denaturation at 95 °C for 3 min, 35 cycles of denaturation for 45 s at 95 °C, annealing for 15 s at 52.5 °C for *ACTIN*, 55.7°C for *AOS*, 58°C for *AOC*, *LOX*, *COI1* and *BBPI*, and extension for 60 s at 72°C. Fluorescence data were collected, and amplicon specificity was confirmed by melting curve analysis and agarose gel electrophoresis. Transcript levels of tested genes were calculated by the double-standard curves method. Three independent biological replicates for each treatment were used for qRT-PCR analyses.

### 2.8. Enzyme Activity Analysis

The PPO activity was quantified using 0.05 M catechol as substrate following the method described byZauberman et al. [41]. The POD activity was determined using the guaiacol method described by Kraus and Fletcher [42]. LOX activity was determined as conjugated diene formation [43]. All these measurements were biologically repeated four times. Leaf samples (0.1 g) were ground in liquid nitrogen and extracted with ice-cold 0.5 M Tris-HCl buffer (1 mL, pH 7.6) and centrifuged at 12,000× *g* for 15 min at 4 °C. The supernatant was maintained at 4 °C until used. The substrate for LOX activity assay was prepared using the method described by Heinisch et al. [44]. The reaction was initiated by mixing 0.2 mL of crude extract with 4.8 mL of the substrate. LOX activity was evaluated by quantifying conjugate diene absorption at 234 nm.

### 2.9. Data Analysis

The SPSS13.0 (SPSS) package for Windows was used for statistical analyses. The difference between treatments and control in bioassays was evaluated using Student t tests (*p* < 0.05); other data were evaluated by one-way ANOVA (*p* < 0.05) with treatment differences among means tested at *p* ≤ 0.05 using a Tukey post hoc test. The data presented are mean ± standard error (*n* = 4).

## 3. Results

### 3.1. Effect on Anti-LF Resistance of Primary Tiller

The body weight increment of LF larvae fed on leaves of rice plants is a solid indicator of plant resistance level. In wild-type plants (Figure 1a), the body weight of the LF larvae fed on the primary tiller from the main stem without any treatment (control) was increased by 44.5%, while those of the LF larvae fed on the primary tiller from the main stem previously damaged by LF and previously treated with MeJA were increased by 24.7% and 31.6% (*p* < 0.01), indicating that LF feeding and MeJA treatment on the main stems improve the antiherbivore resistance of the primary tiller in wild type rice plants.

In OsCOI1 RNAi plants (Figure 1b), the body weights of the LF larvae fed on the primary tiller from main stem without any treatment were increased by 81.4% after 2 days, while the body weights of LF larvae fed on the primary tiller from main stem receiving LF damage and MeJA treatment were increased by 67.1% and 76.3%, respectively, which were 14.3% and 5.2% lower compared to the control (*p* > 0.05).

### 3.2. Effect on JA Levels in Primary Tiller

Upon leaf herbivory by LF larvae, JA levels in the leaves of the primary tiller of WT plants were significantly increased from 3 h to 12 h after LF inoculation (Figure 2a). Compared with the primary tiller from the main stem without any treatment (non-induced + LF), the JA contents in the leaves of the primary tiller of WT plants induced by LF feeding (LF induced + LF) were increased by 69.3%, 69.0%, 21.5%, and 238.2% at 3, 6, 12, and 24 h after LF inoculation (Figure 2a), respectively. After MeJA treatment on the main stem (MeJA induced + LF), the JA contents in the primary tiller leaves of wild-type rice were increased by 40.0%, 54.9%, and 44.1% at 3, 6, and 12 h after LF inoculation, respectively. No induction of JA was found at 48 h after LF inoculation. The results showed that for WT rice plants if the main stem had been infested by LF or treated with MeJA, JA levels in the primary tiller leaves were significantly higher relative to untreated control when attacked by LF.

While for OsCOI1 RNAi plants (Figure 2b), JA levels were relatively lower (Figure 2b) relative to WT plants. LF infestation on the primary tiller only induced JA accumulation at 3 and 6 h after LF inoculation. Upon leaf herbivory, LF pre-treatment on the main stem (LF induced + LF) increased JA levels in the primary tiller leaves by 62.5% and 36.8% at 3 and 6 h after LF inoculation, respectively, compared with untreated control (non-induced + LF). MeJA pretreatment on the main stem (MeJA induced + LF) led to 21.5% increase in JA level only at 3 after LF inoculation. At other time points MeJA pretreatment did not show an induced effect on JA accumulation.

### 3.3. Effect on Trypsin Protease Inhibitor (TrypPI) Levels in Primary Tiller

Protease inhibitors (PIs) play an important role in plant direct defense against lepidopteran (Ryan, 1990; Liu et al., 2021). Trypsin protease inhibitor (TrypPI) levels in the LF-inoculated leaves of the primary tiller of WT plants were enhanced by either LF pre-treatment or MeJA pretreatment on the main stem (Figure 3a). For WT plants, LF pre-treatment on the main stem (LF induced + LF) enhanced TrypPI levels in the primary tiller leaves by 14.3%, 12.7%, 15.1%, and 30.3% at 3, 6, 24, and 48 h after LF inoculation, respectively, relative to untreated control (non-induced + LF). MeJA pre-treatment on the main stem (MeJA induced + LF) enhanced TrypPI levels by 33.9%, 20.3%, 8.0%, and 30.6% at 3, 6, 24, and 48 h after LF inoculation, respectively. In contrast, for OsCOI1 RNAi plants (Figure 3b), either LF pre-inoculation (LF induced + LF) or MeJA pretreatment (MeJA induced + LF) on the main stem did not show any obvious effects on TrypPI levels in the primary tiller leaves relative to untreated control (non-induced + LF). LF infestation itself only induced TrypPI accumulation at 48 h after LF inoculation.

### 3.4. Effect on Transcript Levels of JA Signaling Genes and Bowman−Birk Protease Inhibitor Gene (OsBBPI) in Primary Tiller

After LF pre-inoculation and MeJA pretreatment on the main stem of WT and OsCOI1 RNAi plants for 2 d, the leaves of primary tiller were inoculated and the transcript levels of genes encoding JA biosynthesis and perception, Bowman−Birk protease inhibitor (*OsBBPI*) were examined by real-time qRT-PCR. For WT plants, LF pre-inoculation on the main stem led to increased transcript levels of JA signaling genes encoding allene oxide synthase (*OsAOS*; JA biosynthesis), lipoxygenases (*OsLOX*; JA biosynthesis), allene oxide cyclase (*OsAOC*; JA biosynthesis) and CORONATINE INSENSITIVE1 (*OsCOI1*; JA perception), as well as *OsBBPI*. In wild-type rice, the gene expression of *OsCOI1*, *OsLOX,* and *OsBBPI* in the two treatment groups (LF induced + LF and MeJA induced + LF) was significantly higher at the beginning of LF inoculation relative to two control groups (control and non-induced + LF) (Figure 4a,c,i), indicating the three genes were induced in primary tiller before LF inoculation. LF pre-treatment on the main stem (LF induced + LF) induced expression of *OsLOX* in primary tiller leaves by 2.0- and 5.6-fold at 3 and 12 h after LF inoculation, respectively (Figure 4c). It also induced expression of *OsAOS* in primary tiller leaves by 2.4-, 3.4-, and 1.1-fold at 6, 12, and 24 h after LF inoculation, respectively (Figure 4g).

The expression of *OsBBPI* in the primary tiller leaves of wild-type plants induced by LF feeding on the main stem (LF induced + LF) was significantly higher than that in the control group with untreated main stem (non-induced + LF). It was induced by 16.8-, 10.4-, 20.7-, 8.2-, and 1.7- at 3, 6, 12, 24, and 48 h, respectively. MeJA pretreatment (MeJA induced + LF) on the main stem induced the expression of *OsBBPI* in the primary tiller leaves by 1.7-, 3.8-, 10.2-, 11.3-, and 1.6-fold at 3, 6, 12, 24, and 48 h, respectively.

In OsCOI1 RNAi plants, transcript levels of JA synthesis-related genes *OsCOI1*, *OsLOX,* and *OsAOS* were induced by either LF pre-inoculation or MeJA pre-treatment at the beginning of LF inoculation relative to two control groups (control and non-induced + LF) (Figure 5b,d,f). The transcript level of *OsCOI1* was not affected by LF pre-inoculation and MeJA pre-treatment at other time points tested. The expression of *OsBBPI* in the primary tiller leaves of OsCOI1 RNAi plants was only induced at 6 and 24 h after LF inoculation.

### 3.5. Effect on Activities of Defense-Related Enzymes in Primary Tiller

Defense-related enzymes such as lipoxygenases (LOX), polyphenol oxidase (PPO), and peroxidase (POD) are crucial for plant-induced defense (Ye et al., 2012; Duan et al., 2014). Lipoxygenase (LOX) is a key enzyme involved in JA biosynthesis. The LOX activity was induced by LF infestation in WT plants (Figure 5a). LOX levels in the LF-inoculated leaves of the primary tiller of WT plants were enhanced by LF pre-inoculation on the main stem by 38.6%, 33.3%, 41.4%, 33.6%, 47.5%, and 80.75% at 0, 1, 3, 12, and 24 h, respectively, relative to those without LF pre-inoculation on the main stem (non-induced + LF). While MeJA pretreatment on the main stem only increased LOX levels in the LF-inoculated leaves of the primary tiller by 32.0% and 31.9% at 12 and 48 h, respectively (Figure 5a).

LF pre-inoculation and MeJA pretreatment showed margin effects on PPO activity (Figure 5c). LF pre-inoculation on the main stem increased PPO levels in the primary tiller by 17.5%, 6.2%, 18.3%, and 7.0% at 1, 3, 12, and 48 h, respectively. MeJA pretreatment increased PPO levels by 9.4%, 27.5%, 16.4%, and 7.2% at 1, 3, 12, and 48 h, respectively.

Both LF pre-inoculation and MeJA pretreatment showed strong effects on POD activity (Figure 5c). LF pre-inoculation on the main stem increased POD levels in the primary tiller by 127.7%, 64.8%, 71.7%, and 98.1% at 1, 3, 5, and 12 h, respectively. MeJA pretreatment increased POD levels by 91.0%, 35.1%, 37.9%, 21.8%, and 61.2% at 1, 3, 6, 24, and 48 h after LF inoculation on the primary tiller, respectively.

In OsCOI1 RNAi plants LF pre-inoculation on the main stem enhanced LOX levels in the primary tiller by 37.1%, 97.6%, and 59.1% at 1, 6, and 24 h, and MeJA pretreatment enhanced LOX levels by 47.4%, 62.8%, 60.8%, and 118.4% at 1, 3, 6, and 24 h, respectively (Figure 5b). Either LF pre-inoculation and MeJA pretreatment on the main stem of RNAi plants did not affect levels of PPO and POD in the primary tiller (Figure 5d).

## 4. Discussion

As a huge plant group in nature, clonal plants have great advantages in survival, growth, reproduction, and resource utilization compared with non-clonal plants in most ecosystems. Cloned plants have physiological integration and great ecological significance. Chemicals and nutrients (such as photosynthetic assimilation products, mineral nutrients, water, etc.), can be transferred and shared among individuals in the same clonal plant network. It has been demonstrated that defense signals can be transferred in a clonal network and systemic defense can be induced in undamaged clones after a local attack on one clone [31,32,45]. Gómez and Stuefer found that induced systemic resistance (ISR) was activated by the generalist caterpillar *Spodoptera exigua* and defense signals were transferred among interconnected ramets of the stoloniferous herb *Trifolium repens* [32]. Gómez et al. showed that systemic defense induction in one ramet of *T. repens* greatly reduced herbivore damage by *S. exigua* on young ramets, but slightly enhanced mature ramets [33]. Meanwhile, defense induction also led to increased leaf strength and thickness, decreased leaf soluble carbohydrates, and substantially changed phenolic composition in undamaged individuals connected to attacked ramets. However, this phenomenon has not been confirmed in tillering plants and important crops. Our results in this study show that defense signals were shared and transferred between the main stem and its primary tillers after an herbivore attack on the main stem. Such warning signals from the attacked main stem induced stronger defense responses in primary tillers. Both LF attack and MeJA treatment on the main stem enhanced the antiherbivore resistance level of primary tillers, leading to poorer growth of LF larvae on the tillers (Figure 1a).

Induced antiherbivore defense is an important function of the plant immune system [1]. Upon attack by insect herbivores plants can initiate an array of physiological, biochemical, and morphological responses [46]. To further determine the possible mechanism involved, we examined the defense responses in the primary tillers after LF leaf herbivory and MeJA treatment on the main stem. These defense responses included JA contents, protease inhibitor level and transcript levels of JA signaling genes and protease inhibitor gene, and defense-related enzyme activities in the primary tillers.

Protease inhibitors in plants impose a major food quality restriction on insect herbivores [47,48]. Our results show that either LF pre-treatment or MeJA pretreatment on the main stem enhanced TrypPI levels in the leaves of primary tillers (Figure 3a). Meanwhile, LF pre-treatment on the main stem induced the expression of *OsBBPI,* a PI synthesis gene, in the primary tiller leaves by 16.8-, 10.4-, 20.7-, and 8.2-fold at 3, 6, 12, and 24, respectively; and MeJA pretreatment induced the expression of *OsBBPI* by 3.8-, 10.2-, and 11.3-fold at 6, 12, and 24 h, respectively (Figure 3a). Strong induction of the TrypPI level and transcripts of *OsBBPI* may provide rice protection against LF attack [49]. To determine the role of JA signaling in the communication between stem and primary tillers, the transgenic rice plants OsCOI1-RNAi deficient in JA perception were used to investigate resulting changes in anti-LF resistance and defense responses in the primary tillers. LF leaf herbivory and MeJA treatment on the main stem led to a higher accumulation of JA in primary tillers in both WT plants and COI RNAi plants (Figure 2a,b), suggesting activation of JA signaling in primary tillers upon insect herbivory on the main stem. Real-time RT-qPCR analysis showed that LF leaf herbivory and MeJA treatment on the main stem strongly induced transcription of genes encoding JA biosynthesis (*OsLOX*, *OsAOS*, *OsAOC*) and perception (*OsCOI1*) (Figure 4). Compared with non-LF inoculation on the main stem (non-induced + LF), LF leaf herbivory on the main stem was induced earlier and stronger transcription of *OsCOI1*, *OsLOX*, *OsAOS,* and *OsAOC*. In contrast, in *COI1* RNAi plants, defense responses induced by main stem LF infestation in primary tillers were either devoid or minor, indicating *COI1* is an essential component for the regulation of JA-mediated defense communication. In LF-attacked primary tillers, the strong activation of the JA pathway in WT plants and compromised induction of defense responses, and induced anti-LF resistance in *COI1* RNAi plants suggest that JA signaling plays a crucial role in defense communication between the main stem and primary tillers.

In the plant antiherbivore defense system, LOX, PPO, and POD enzymes are important components. LOX enzyme is a key enzyme of JA biosynthesis, which catalyzes the peroxidation of polyunsaturated fatty acids. It has been well-documented that LOX activity plays a vital role in plant antiherbivore defense [50,51]. PPO catalyzes the oxidation of phenolic compounds into highly reactive quinones with antiherbivore activity [52]. Overexpression of PPO enhanced tomato resistance to common cutworms [53]. The results of this study showed that after the stem had been attacked by LF larvae or treated with signaling compound MeJA, the activities of LOX, PPO, and POD in the leaves of the LF-infested primary tillers were significantly higher than those in the controls (Figure 5a). After silencing the *OsCOI1* gene, up insect herbivory by LF larvae, LF pre-inoculation and MeJA pretreatment on the stem did not show obvious effects on PPO and POD activities in the primary tillers, but the two treatments still showed an induced effect on LOX activity.

Our findings show that there exists a defense signal communication network among rice clonal plants. When the main stem is attacked by insect pests, rice plants will initiate the defense signal communication with tiller plants, and activate the antiherbivore defense responses in the clonal network, leading to enhanced antiherbivore resistance in primary tillers connected to the main stem. Although we did not analyze the defense responses in the secondary and tertiary tillers, we speculate they have been affected and induced by the main stem insect attack. This study revealed the defense communication among different clonal ramets of rice and the induced defense signal transmission in the clonal plant network. The findings have important ecological and evolutionary implications for clonal integration in defense signal sharing after localized herbivore attacks in major food crops and may provide a theoretical basis for ecological control of rice pests using a systemic defense of the clonal plant.

In summary, this work demonstrates that systemic antiherbivore defense operates in the clonal network of rice plants. When the stem is attacked by Lepidoptera pest LF, the primary tillers strongly enhance their induced defense responses: higher levels of JA, protease inhibitor, transcripts of defense-related genes, and defense-related enzyme activities, leading to enhanced antiherbivore resistance against LF. We also demonstrate that JA signaling plays a crucial role in mediating defense communication between the main stem and tillers in rice plants. We propose that systemic defense is an indispensable part of physiological integration and extensively operates in clonal plant networks.

## Figures and Tables

**Figure 1 plants-12-01199-f001:**
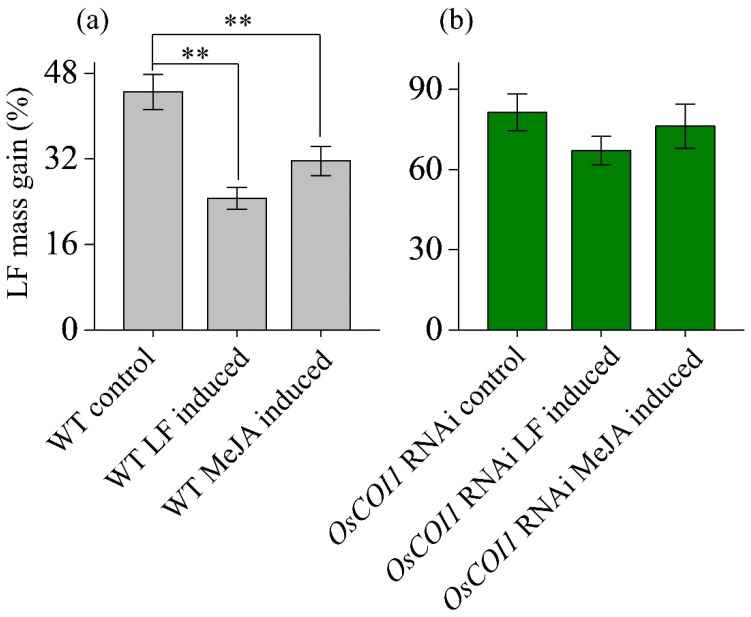
Mass gains of leaf folder (LF) larvae fed on primary tillers of wild-type (**a**) and *OsCOI1* RNAi (**b**) rice plants after the main stems had been induced by LF feeding or MeJA (1 mM) treatment for 2 d. Values are mean ± standard error (*n* = 20). Asterisks above bars indicate significant differences of treatment compared to the control (** *p* < 0.01 according to Student’s *t*-test).

**Figure 2 plants-12-01199-f002:**
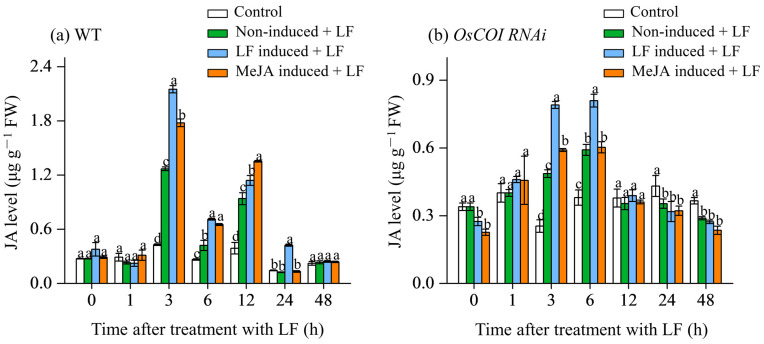
JA levels in the leaves of primary tiller of wild-type (**a**) and *OsCOI1* RNAi (**b**) rice plants after main stem had been induced by LF feeding or MeJA treatment for 2 d. Each phenotype received four treatments: control, both main stem and primary tiller did not receive any treatment; non-induced + LF, main stem did not receive any treatment, but primary tiller received LF infestation; LF induced + LF, main stem was inoculated with LF larvae for two days and then the primary tiller was inoculated with LF larvae; MeJA induced + LF, main stem was sprayed with MeJA for two days and then the primary tiller was inoculated with LF larvae. Values are mean ± standard error (*n* = 4). For each time point, letters above bars indicate significant differences among treatments (*p* < 0.05 according to Tukey’s multiple range test).

**Figure 3 plants-12-01199-f003:**
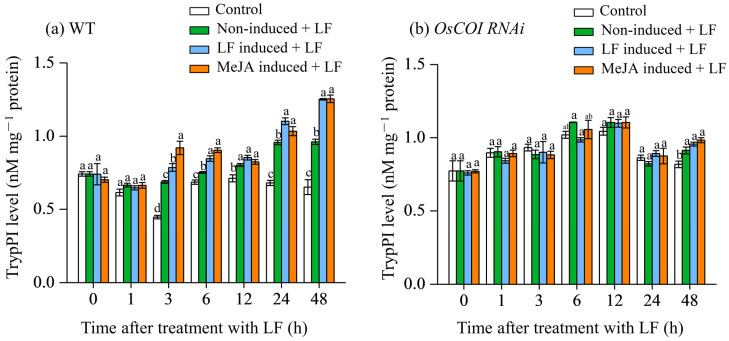
Trypsin protease inhibitor (TrypPI) levels in the leaves of primary tiller of wild-type (**a**) and *OsCOI1* RNAi (**b**) rice plants after main stem had been induced by LF feeding or MeJA treatment for 2 d. The treatments shown were performed as described in Figure 2. Values are mean ± standard error (*n* = 4). For each time point, letters above bars indicate significant differences among treatments (*p* < 0.05 according to Tukey’s multiple range test).

**Figure 4 plants-12-01199-f004:**
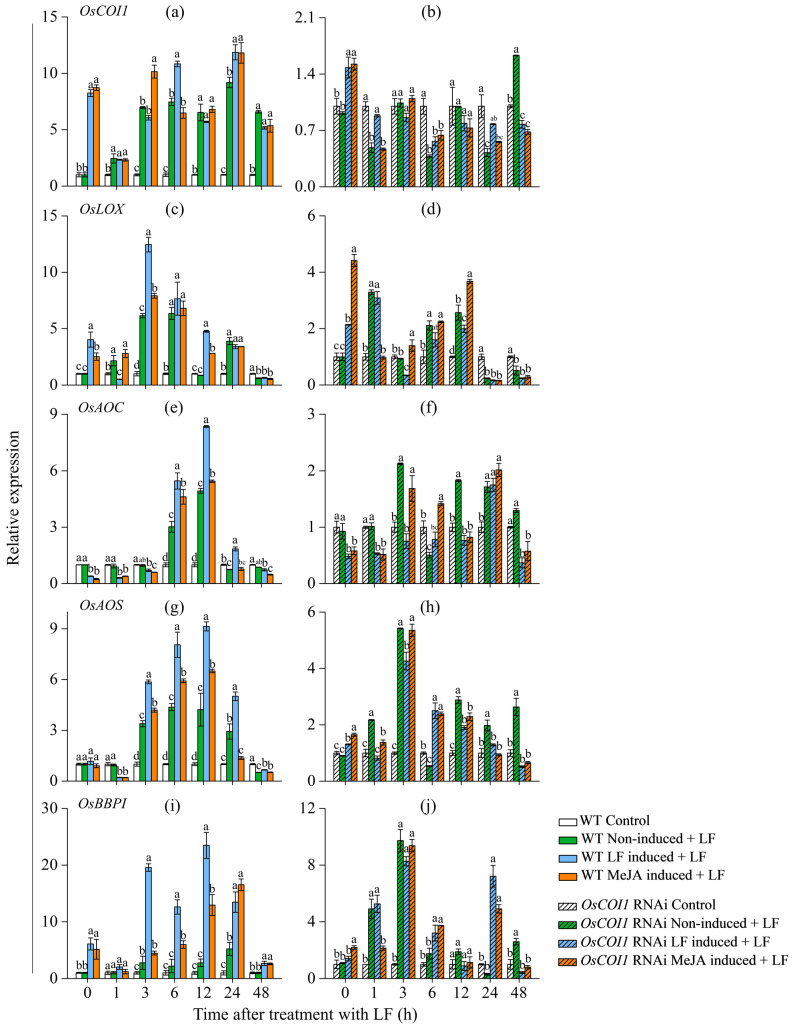
Transcript levels of *OsCOI1, OsLOX, OsAOC, OsAOS,* and *OsBBPI* in the leaves of primary tiller of wild-type (**a**,**c**,**e**,**g**,**i**) and *OsCOI1* RNAi (**b**,**d**,**f**,**h**,**j**) rice plants after main stem had been induced by LF feeding or MeJA treatment for 2 d. The treatments shown were performed as described in Figure 2. Values are mean ± standard error (*n* = 4). For each time point, letters above bars indicate significant differences among treatments (*p* < 0.05 according to Tukey’s multiple range test).

**Figure 5 plants-12-01199-f005:**
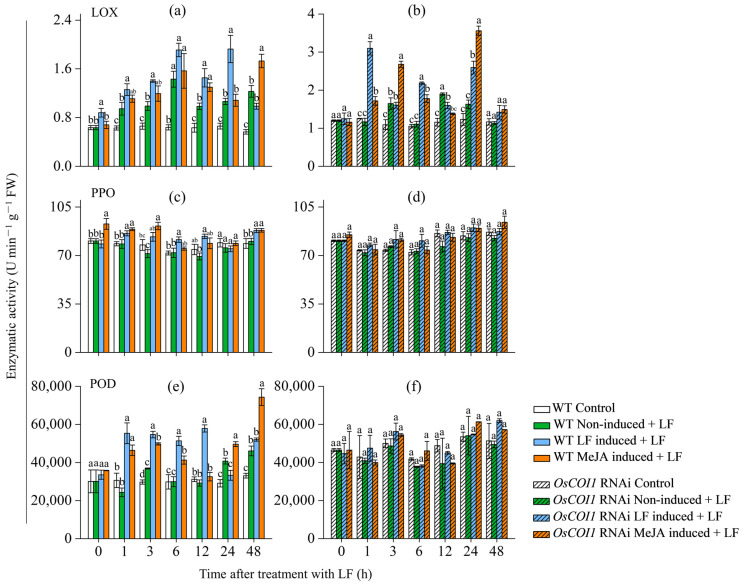
Enzymatic activities of LOX, PPO, and POD in the leaves of primary tiller of wild-type (**a**,**c**,**e**) and *OsCOI1* RNAi (**b**,**d**,**f**) rice plants after main stem had been induced by LF feeding or MeJA treatment for 2 d. The treatments shown were performed as described in Figure 2. Values are mean ± standard error (*n* = 4). For each time point, letters above bars indicate significant differences among treatments (*p* < 0.05 according to Tukey’s multiple range test).

## Data Availability

All the data analyzed during this study have been included in this article.

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
