# Peer review of "Insect Herbivory on Main Stem Enhances Induced Defense of Primary Tillers in Rice (*Oryza sativa* L.)"

_plants, 2023, doi:10.3390/plants12051199_

Round 1
Reviewer 1 Report
Dear Tong et al.
Your manuscript on the role of main stem defense communication to clonal tiller plants after C. medinalis infestation is very thorough and intriguing. I think this is a very important addition to the area of plant defense signaling, especially for clonal plants like rice. In this manuscript you have demonstrated that C. medinalis infestation and JA treatment of the main tiller enhances plant resistance to subsequent infestation. Furthermore, LF infestation of the main stem and JA treated increased JA accumulation and related defensive enzymes in the main tiller. I think it was especially compelling that your group used RNAi to render the plants insensitive to JA and observed that in fact JA does regulate much of the defense signaling between main stems and tillers. Overall, this work was very thorough and organized and tells a wonderful story.
There were a few areas of the manuscript that I think could be adjusted to improve it. Overall, some of the grammar needs to be addressed. Some examples are:
L161 and 164 - "weighted" should be "weighed"
L262 - a comma after respectively
L326 - use of hyphens, such as "JA synthesis-related"
L340 - redundant "related" in "defense-related related"
Materials and Methods
- How were the field-collected rice leaffolder caterpillars confirmed to species by a specific manner?
Results
- In Fig 1. the LF mass gain is much higher in (b) that (a). Is there a reason for this or just natural variation between bioassays?
Thank you for the joy of reading this fun manuscript. I wish you the best of luck in the future.
Best wishes,
Hillary
Reviewer 2 Report
In this intriguing study, the role of systemic signalling and clonal integration in plant defence against herbivores is discussed. The research team employed rice and its pest, rice leaffolder, to examine how the main stem and clonal tillers communicate antiherbivore responses. My queries and suggestions for improvement are listed below:
1. If the observed systemic induction of antiherbivore defence in clonal tillers is a phenomenon widespread across several plant species or specific to rice plants, that is one crucial question to ask about this study. That must be reflected in the introduction and discussion with appropriate citations and crop-specific examples.
2. In addition, it would be intriguing to find out if signalling pathways other than JA are involved in facilitating the transmission of antiherbivore defences in clonal plant networks. Please comment.
3. Please incorporate more scientific outcomes in the abstract part. Additionally, remove the error in line 17. Write a future prospect sentence in the abstract. How this study will assist the management approaches?
4. Line 94-98 please provide a citation.
5. This sentence “We hypothesized that leaf herbivory on main stem will increase antiherbivore resistance and defense responses in primary tillers and JA signaling plays a crucial role in defense communication between the main stem and its primary tillers.” It is a finding-driven hypothesis focusing on only one factor “JA” please modify the sentence.
6. Several typographical errors are evident in material and methods.
7. Line 167: modify the caption
8. Figure 1 legend requires modification
9. Figure 2b, 3b the bar should be modified for better visibility.
Round 2
Reviewer 2 Report
All the queries and suggestions were addressed correctly.